# Community health worker-delivered counselling for common mental disorders among chronic disease patients in South Africa: a feasibility study

Bronwyn Myers,[1] Petal Petersen-Williams,[1,2] Claire van der Westhuizen,[3] Crick Lund,[3,4] Carl Lombard,[5] John A Joska,[2] Naomi S Levitt,[6] Christopher Butler,[7] Tracey Naledi,[8] Peter Milligan,[2] Dan J Stein,[2,9] Katherine Sorsdahl[3]

For numbered affiliations see end of article.

**Correspondence to**
Dr Bronwyn Myers;
bmyers@mrc.ac.za

## ABSTRACT

**Objectives** To examine the feasibility and acceptability of integrating a 'designated' approach to community health worker (CHW)-delivered mental health counselling (where existing CHWs deliver counselling in addition to usual duties) and a 'dedicated' approach (where additional CHWs have the sole responsibility of delivering mental health counselling) into chronic disease care.

**Design** A feasibility test of a designated and dedicated approach to CHW-delivered counselling and qualitative interviews of CHWs delivering the counselling.

**Setting** Four primary healthcare clinics in the Western Cape, South Africa allocated to either a designated or dedicated approach and stratified by urban/rural status.

**Participants** Forty chronic disease patients (20 with HIV, 20 with diabetes) reporting hazardous alcohol use or depression. Interviews with seven CHWs.

**Intervention** Three sessions of structured mental health counselling.

**Main outcome measures** We assessed feasibility by examining the proportion of patients who were willing to be screened, met inclusion criteria, provided consent, completed counselling and were retained in the study. Acceptability of these delivery approaches was assessed through qualitative interviews of CHWs.

**Results** Regardless of approach, a fair proportion (67%) of eligible patients were willing to receive mental health counselling. Patients who screened positive for depression were more likely to be interested in counselling than those with hazardous alcohol only. Retention in counselling (85%) and the study (90%) was good and did not differ by approach. Both dedicated and designated CHWs viewed the counselling package as highly acceptable but requested additional training and support to facilitate implementation.

**Conclusions** Dedicated and designated approaches to CHW-delivered mental health counselling were matched in terms of their feasibility and acceptability. A comparative efficacy trial of these approaches is justified, with some adjustments to the training and implementation protocols to provide further support to CHWs.

## BACKGROUND

In South Africa, like other low/middle-income countries (LMIC), there is a substantial treatment gap for common mental disorders (CMDs).[1] Untreated mental disorders contribute to the country's large burden of disease associated with other non-communicable diseases (NCDs) such as diabetes and chronic communicable diseases (such as HIV).[2] South African studies have demonstrated high levels of CMD and other NCD multimorbidity among the general population[3 4] and among patients attending primary healthcare (PHC) facilities.[5] As untreated CMDs are associated with poorer adherence to chronic disease treatment and more adverse outcomes,[6 7] there is a public health imperative to reduce the treatment gap for these patients.

A recommended strategy for reducing this gap is the integration of counselling for CMDs into chronic disease services provided in PHC clinics.[8] Severe shortages of mental health specialists in South Africa have impacted on the implementation of this strategy.[9] To overcome this challenge,

and in keeping with WHO's recommendations for increasing mental healthcare access,[10] South Africa has endorsed task sharing of basic mental health counselling to non-specialist providers, including community health workers (CHWs) deployed within PHC services.[11] Systematic reviews highlight the feasibility and acceptability of using trained CHWs to deliver counselling in LMICs.[12–14] There is also emerging evidence suggesting that CHW-delivered interventions may reduce both depression and hazardous alcohol use.[15] Despite this promising evidence, uncertainty about how best to configure resources within the PHC system to enable CHW-delivered mental health interventions within chronic disease care has delayed implementation. Some argue that CHWs working in chronic disease teams have spare capacity and can be *designated* to deliver counselling in addition to their usual responsibilities.[16] Others contend that these CHWs are already overloaded so it is impossible for them to deliver additional counselling. In this view, CHWs *dedicated* to the delivery of mental healthcare are needed to ensure the feasibility of this service.[11]

### Strengths and limitations of the study

► The first study to compare the feasibility and acceptability of two different approaches to resourcing community health worker (CHW)-delivered mental health counselling.
► The use of mixed methods enables triangulation of findings, increasing confidence in the results.
► Qualitative interviews provide a rich description of CHWs' experiences of delivering mental health counselling.
► There were only a small number of clusters; therefore, we cannot draw inferences from the quantitative data.
► CHWs included in this study may not be representative of the total population of CHWs working in chronic disease care.

To guide health planners in their decisions about how to integrate mental health counselling into chronic disease care, the feasibility, acceptability and cost-effectiveness of these approaches to CHW-delivered counselling must be established. This study examined the feasibility of integrating a dedicated and a designated approach to CHW-delivered counselling for CMDs into chronic disease care in PHC facilities in the Western Cape province of South Africa. Specific aims were to explore (1) the feasibility of recruiting and retaining patients with chronic disease for CHW-delivered mental health counselling and (2) dedicated and designated CHWs' perceptions of the feasibility and acceptability of delivering mental health counselling to patients with chronic disease. Findings will inform patient recruitment and retention protocols and CHW training and intervention protocols that will be used in a future trial examining the relative effectiveness and cost-effectiveness of these two approaches to CHW-delivered counselling.

## METHODS

This manuscript is in accordance with the 'Good reporting of a mixed methods study guidelines' (GRAMMS). See online supplementary file 1 for the GRAMM checklist. The study, conducted from May to October 2016, comprised a feasibility test of CHW-delivered counselling and qualitative interviews of CHWs.

### Study sites, participants and procedures

We recruited patients with chronic disease from four PHC clinics in the Western Cape. Two sites (stratified by urban/rural status) used *dedicated* CHWs and the remaining two sites (stratified by urban/rural status) used *designated* CHWs to deliver the counselling. At the designated sites, the job descriptions of CHWs employed by non-governmental organisations (NGOs) to deliver

HIV adherence counselling within these facilities were expanded to include mental health counselling. At the dedicated sites, additional CHWs were appointed with the sole responsibility of delivering this new service.

During the recruitment period, health providers asked all patients presenting for HIV or diabetes treatment about their past-year alcohol use and recent low mood. Patients who responded positively were referred to a study assessor who requested verbal consent for study eligibility screening. Eligibility criteria included being (1) at least 18 years old; (2) on antiretroviral therapy (ART) for HIV or medication for diabetes; and (iii) reporting hazardous/harmful drinking using the Alcohol Use Disorders Identification Test (AUDIT)[17] or probable depression using the Center for Epidemiological Studies Depression scale (CES-D).[18] AUDIT has been validated for use in South Africa, with cut-off scores ≥8 indicating hazardous alcohol use.[19] CES-D measures change in symptoms of depression, with a cut-off score ≥16 indicating probable depression.[20] Patients receiving other mental health treatment or participating in another study were excluded. We followed these procedures until we recruited 10 participants per site (five unique participants with HIV and five unique participants with diabetes) for a total sample of 40 participants.

At the enrolment appointment, the assessor obtained informed consent before administering the baseline assessment in English, Afrikaans or isiXhosa (main languages of the region). This computer-assisted assessment collected sociodemographic information on age, race, gender, education level and employment status; HIV and diabetes status; and used the AUDIT and CES-D to assess extent of hazardous/harmful alcohol use and depression. After completing this assessment, three counselling sessions (spaced at least a week apart) were scheduled with the CHW. Participants had 6 weeks to complete all three sessions. Participants returned for a follow-up appointment 1 month after their last counselling session. At this appointment, the assessor readministered the baseline assessment. All study activities occurred in private rooms at the PHC facility. Participants received grocery vouchers for completing each research assessment; they were not incentivised to attending counselling.

After this feasibility test, in order to better understand and corroborate the quantitative findings, an independent qualitative researcher interviewed the seven CHWs who delivered the intervention. Interviews were conducted in English, were audio-recorded and lasted up to 60 min. Interviews followed a semistructured guide with opening questions and follow-up probes to elicit CHWs' experiences of delivering the intervention, barriers to delivery and suggestions for altering the proposed training and intervention protocols to enhance feasibility and acceptability.

### Description of counselling programme and CHWs

Counselling comprised three structured sessions of motivational interviewing (MI) and problem-solving therapy

(PST) which has evidence for efficacy among South African PHC patients.[21] The rationale for selecting this approach has been described elsewhere.[22] During this programme, CHW and participant collaborated to identify and explore problems within the participant's life while the CHW taught the participant a structured PST approach to resolving these concerns. Participants learnt strategies for addressing problems that are important and resolvable, for dealing with negative and intrusive worries that are unrelated to their life goals and for coping with important problems that are unresolvable. Participants rehearsed these new skills through exercises and take-home activities contained in a patient handbook.

Dedicated and designated CHWs who were selected and trained to deliver the intervention were matched on qualifications (completion of high school and training as HIV adherence counsellors), experience as HIV adherence counsellors and remuneration. Both the dedicated and designated CHWs received 3 days of training in screening for hazardous/harmful alcohol use and depression, the counselling programme and referral pathways. All counselling sessions were audiotaped; a registered psychological counsellor reviewed a random sample of these for fidelity using a simple fidelity checklist. No differences in fidelity were observed between the designated and dedicated CHWs. Supervision and debriefing of all the CHWs in the dedicated and designated arms was task shared from a psychologist to a registered psychological counsellor, in line with the South African National Mental Health Policy Frameworks vison of district mental health teams.[11] This psychological counsellor provided both dedicated and designated CHWs with weekly individual supervision in which feedback on counselling sessions and how to improve fidelity and quality of counselling, retraining in aspects of the programme occurred if needed, and debriefing was provided for difficult or challenging cases.

### Patient and public involvement
Patients with chronic disease and the broader public were involved in the design and implementation of this study. The two task-sharing models and the design of the feasibility test were informed by interviews with patients with chronic disease with untreated depression or hazardous alcohol use.[23] Our stakeholder advisory group that comprises representatives from the Department of Health, PHC facilities, charities and service user organisations proposed this feasibility test and contributed to the design of the study and the interpretation of findings.

### ANALYSES
SPSS V.25.0 was used to assess the proportion of patients who (1) were willing to be screened, (2) met inclusion criteria, (3) enrolled into counselling, (4) completed counselling and (5) were retained in the study. Possible differences in performance on recruitment and retention indicators by site (urban versus rural; dedicated versus

designated) and patient characteristics were explored using $\chi^2$ tests for categorical and t-tests for continuous variables. All testing was two-sided and used a significance level of 0.05.

We used the framework approach[24] to analyse qualitative data. Two study staff used NVivo V.11 to code interview transcripts; they met regularly to compare notes and resolve discrepancies. A third person was not needed to break coding ties. No new codes emerged after coding half the transcripts, implying thematic saturation. Intercoder reliability was high, with a kappa score of 0.92.

### RESULTS
#### Feasibility of recruitment and retention
Of the 553 chronic disease patients screened for recent alcohol use and depressed mood, 262 (48%) were potentially eligible for study inclusion and referred for eligibility screening. About a quarter (26%, n=69) declined screening, mainly due to lack of time or interest. There were no demographic and site differences between those that accepted and those that declined the offer of screening.

Of the 193 remaining patients, 101 met inclusion criteria. Sixteen patients (16%) were eligible on their AUDIT scores, 69 (58%) on their CES-D scores and 16 (16%) on their AUDIT and CES-D scores. Sixty-seven (66%) of these eligible patients were interested in participation. Figure 1 presents reasons for declining participation. Site characteristics were not associated with declining participation. Patients who were eligible based on their CES-D scores were more likely to be interested in participation (74%) than those who were only eligible based on their AUDIT scores (33%; p=0.005). Gender was associated with interest in participation: 74% of eligible women versus 46% of eligible men (p=0.018). Only 40 of these 67 patients returned for their enrolment visit; the remainder were untraceable. Participants who were untraceable were more likely to be recruited from urban than rural sites (51% vs 26%; p=0.038). There were no other differences between eligible, interested patients who were enrolled and those who were not.

Of these 40 participants, 22 were HIV positive and 18 had diabetes (two had both conditions). Almost all participants (n=38, 95%) had CES-D scores ≥16 while only 20% of the sample (n=8) met criteria for hazardous/harmful alcohol use. Table 1 depicts the demographic and clinical characteristics of the sample. Of these 40 participants, 34 (85%) completed the entire counselling programme (figure 1 shows reasons for not completing counselling). Treatment completers and non-completers did not differ on demographic or clinical characteristics. Participants from rural sites were more likely to complete treatment than those from urban sites (95% vs 75%; p=0.04). Participants from dedicated and designated sites were equally likely to complete treatment. Thirty-six (90%) participants completed the follow-up assessment. Figure 1 provides reasons for attrition. Study completers and

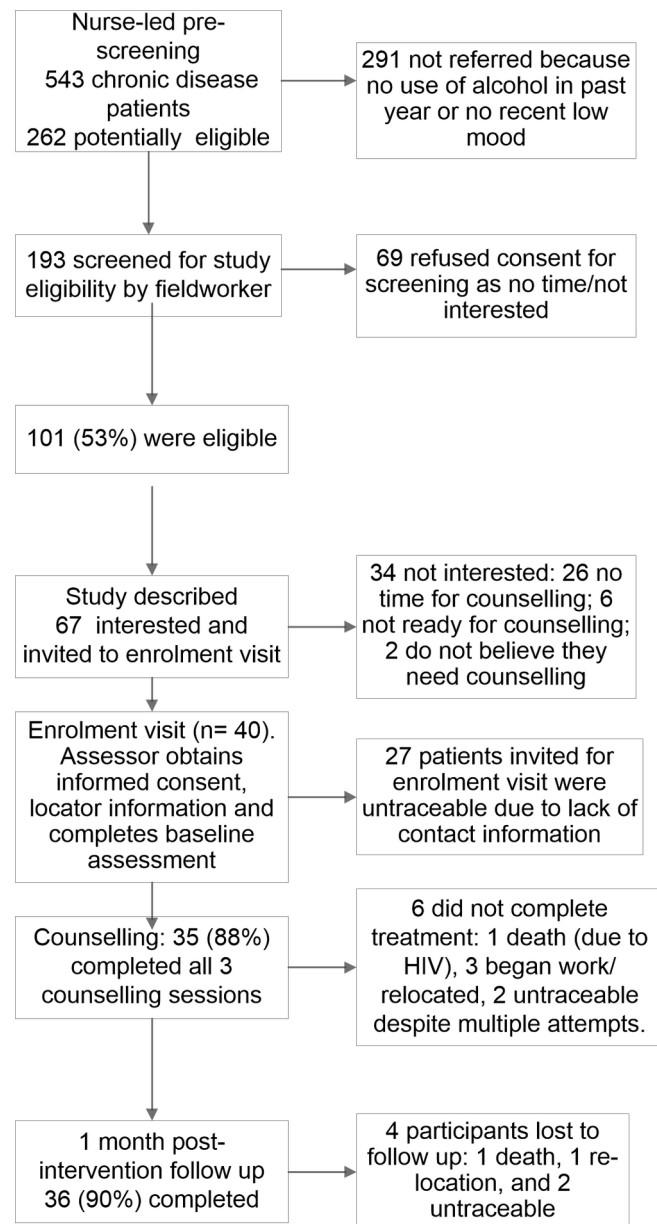

**Figure 1** Patient flow diagram.

non-completers did not differ on any demographic or clinical characteristics. Site characteristics were not associated with study completion.

### Perceptions of the feasibility and acceptability of the counselling programme

Three themes emerged from the data that reflect CHWs' perceptions of the feasibility and acceptability of implementing the proposed counselling programme. The first theme describes CHWs' perceived confidence in their ability to deliver this new service. The second theme describes the acceptability of the proposed counselling package to CHWs. The third theme describes CHWs' experiences of barriers to counselling delivery and their recommendations for mitigating these barriers.

### Confidence in ability to deliver mental health counselling

Both dedicated and designated CHWs had limited experience in delivering psychological counselling; this programme represented an expansion of their scope of work. They described the intervention as 'something new' and 'different from the counselling we did'. As this was a shift in practice, they had some concerns about whether the training protocols adequately prepared them for intervention delivery. Most CHWs thought additional time was needed to rehearse the content of the sessions and to build their confidence and competency:

> The training should be about five days. Five days gives you enough time to role play and ask questions and grasp everything … The role play, there was not actually proper time for that.
>
> (dedicated CHW)

While most CHWs reported initially being a 'bit scared because it was something new', they still viewed their involvement with the programme as an opportunity to learn how to work more effectively with their patients and 'to develop their skills':

> In the past I was running out of options because they [patients] were depressed … I learned through this intervention to give the participant space to open up.
>
> (designated CHW)

### Perceived benefits of the intervention

All CHWs viewed the intervention as acceptable and beneficial to patients with chronic disease. Dedicated and designated CHWs emphasised how 'there is really a need for this intervention' and that 'a lot of patients here can benefit.' The acceptability of the programme to patients seemed high, with CHWs commenting on how well patients engaged with the intervention. As one dedicated CHW described, 'It seems like when I'm doing session one, they can't wait for session two to give me feedback'. All CHWs were able to provide concrete examples of how patients applied the problem-solving skills they had learnt to resolve problems. They also observed positive changes in patients, which was personally rewarding:

> I saw that each session is making a difference to other people. And to see that people are coming again to finish their sessions … it's telling me the intervention is making a difference to people.
>
> (designated CHW)

Nonetheless, some CHWs felt that patients could benefit from more sessions for there to be 'a difference in a person's life.' Some CHWs suggested offering an additional counselling session to review the patient's progress in reaching their goals.

Not only were there perceived benefits for patients who received this intervention, but other patients also seemed to benefit from the designated CHWs' enhanced counselling skills. Several designated CHWs noted how they

**Table 1** Baseline demographic and clinical characteristics of the overall sample (n=40) and by chronic condition subgroup

| Variable | Overall sample N (%) | Treatment for HIV | | | Treatment for diabetes | | |
|---|---|---|---|---|---|---|---|
| | | Yes (n=22) N (%) | No (n=18) N (%) | P value | Yes (n=20) N (%) | No (n=20) N (%) | P value |
| Gender: Female | 34 (85.0%) | 18 (81.8) | 16 (88.9) | 0.673 | 18 (90.0) | 16 (80.0) | 0.661 |
| Race: Mixed race ancestry | 22 (55.0%) | 8 (36.4) | 14 (77.8) | 0.012* | 14 (70.0) | 8 (40.0) | 0.111 |
| Education | | | | 0.408 | | | 0.337 |
| Some primary | 12 (30.0) | 7 (31.8) | 5 (27.8) | | 7 (35.0) | 5 (25.0) | |
| Some secondary | 23 (57.5) | 11 (50.0) | 12 (66.7) | | 12 (60.0) | 11 (55.0) | |
| Completed secondary | 5 (12.5) | 4 (18.2) | 1 (5.6) | | 1 (5.0) | 4 (20.0) | |
| Employment (yes) | 14 (35.0) | 10 (45.5) | 4 (22.2) | 0.186 | 5 (25.0) | 9 (45.0) | 0.320 |
| Age in years (M, SD) | 44 (12.2) | 38.9 (9.7) | 50.3 (12.2) | 0.002* | 50.0 (11.8) | 38.1 (9.5) | 0.001* |
| AUDIT ≥8 | 8 (20.0) | 6 (27.3) | 2 (11.1) | 0.258 | 2 (10.0) | 6 (30.0) | 0.235 |
| CES-D ≥16 | 38 (95.0) | 20 (90.9) | 18 (100) | 0.492 | 20 (100) | 18 (90.0) | 0.487 |

*Significant association at p<0.05.
AUDIT, Alcohol Use Disorders Identification Test; CES-D, Center for Epidemiological Studies Depression.

now routinely used their new counselling skills in their interactions with other patients:

> I put it as part of my daily work … when I am busy with a session with another patient, then I also try to bring in Project Mind problem solving. It helps me understand where my patient is coming from.
>
> (designated CHW)

In addition, both the dedicated and designated CHWs appeared to benefit personally from this programme, describing how they now applied PST strategies to resolve problems in their own lives and limit negative thoughts:

> I used to worry a lot. when I was doing the intervention then I realised that there are things we worry about that are not important. … so it seems like I am helping someone else, and I am also helping myself.
>
> (dedicated CHW)

### Lessons for future implementation

According to CHWs, support from the PHC facility, other CHWs and the NGOs that employed them (where applicable) was critical for facilitating counselling delivery. In facilities where the designated and dedicated CHWs described facility staff as 'supportive' and 'interested', the counselling proceeded smoothly. However, in PHC facilities where staff seemed less interested, CHWs were often interrupted during counselling, impacting on the therapeutic alliance. For these facilities, CHWs recommended that facility managers and staff are thoroughly informed about the initiative so that 'they know what it is all about so when you are busy with a client they just give you some time to do it.'

For designated CHWs, competing priorities and limited time were barriers to counselling delivery. These CHWs felt they were 'doing three jobs', with almost all mentioning that it would be easier to deliver the programme if all they

did 'was concentrate on the intervention.' This was not an issue for dedicated CHWs who had more time available to conduct the counselling. Where fellow CHWs assisted the designated CHW with their usual HIV care responsibilities, thereby freeing up some of the designated CHW's time, the CHW felt more empowered to deliver the intervention. This realignment of the responsibilities of the CHW team to accommodate mental health counselling happened organically at some (but not all) of the designated sites. To facilitate this realignment in the future, designated CHWs thought it would be helpful for NGO employers and supervisors to be more involved in training and discussions about the implementation of the intervention, so that they 'have a better understanding' about what is required from their staff:

> There was some miscommunication because they [NGO] didn't have a lot of understanding as to what was expected from us. So for future, it would be nice if the managers could be in one of the training sessions just to know what it is all about … to avoid confusion as to the time spent with patients.
>
> (designated CHW)

### DISCUSSION

This study examined the feasibility and acceptability of a dedicated or designated approach to CHW-delivered mental health counselling within the context of chronic disease care in South Africa and found few differences. It contributes to the growing evidence base of the feasibility and acceptability of task-shared mental health interventions in Africa and other LMICs.[25–27] Findings suggest that regardless of whether counselling was delivered by a dedicated or designated CHW, (1) a fair proportion of patients were willing to receive mental health counselling; (2) retention in counselling and the study was

good; and (3) both dedicated and designated CHWs viewed the counselling package as highly acceptable but requested additional training and support to facilitate implementation.

More specifically, findings suggest high levels of unmet need for and adequate uptake of mental health counselling among patients with chronic disease in this setting. This supports the feasibility of recruiting patients with chronic disease for mental health counselling for a larger study. However, the counselling refusal rate was higher than anticipated—particularly among hazardous drinkers without depression and men. Counselling refusals rates were similar for the dedicated and designated models. As we approached patients who were not actively seeking mental healthcare, some may have been ambivalent about the offer of counselling. Concern about health provider stigma towards people with alcohol problems also may have contributed to this finding.[28] Further, as psychological distress is a known driver of counselling readiness among patients with alcohol problems,[29] patients who reported hazardous alcohol use without co-occurring depression may have been less motivated to initiate counselling than those with depression. Guided by these findings, we have modified our recruitment protocols to ensure a future trial is able to recruit sufficient numbers of people with hazardous alcohol use and depression to allow for the assessment of change on either outcome. Recruitment protocols now include health talks at PHC facilities to help reduce the stigma associated with CMDs; distribution of handouts that provide patients with information about the study before they are approached for screening; and additional opportunities for eligible patients who decline the initial offer of counselling to receive counselling.

Our finding of men being less interested in screening and counselling is similar in other parts of Africa,[26] and in keeping with the generally poor rates of healthcare utilisation by men.[30] All the CHWs in this pilot test were women which, along with views that clinics are places for women and children,[30] may have contributed to men's reluctance to accept the offer of counselling. To enhance the uptake of mental health counselling among men, more work needs to be done to understand men's counselling preferences and barriers to health service utilisation. Consideration should also be given to actively seeking to appoint men in the dedicated counsellor role.

Findings also show the feasibility of retaining patients with chronic disease in CHW-delivered mental health counselling, regardless of whether a dedicated or designated approach is used. However, counselling completion rates were better for the rural sites—possibly because some participants from urban sites gained employment and were unable to attend facility-based counselling. To overcome this challenge, we have adjusted our protocols to allow CHWs to deliver counselling via telephone. Nonetheless, taken together, the high counselling completion rate, feedback about patient engagement and requests for additional sessions suggest that patients

found the counselling acceptable and beneficial. Given these requests, we have decided to add an optional problem-solving session to our counselling protocol, which we anticipate may enhance and maintain treatment gains.

Both dedicated and designated CHWs generally thought the counselling was feasible to implement. Initially, many CHWs had reservations about delivering mental health counselling, but these reservations dissipated with training, delivery experience and supportive supervision. CHWs' observation that counselling improved participants' well-being reinforced their views of programme acceptability. Designated CHWs also noted how these additional counselling skills improved their interactions with other patients that they provided with chronic disease care; augmenting their positive views of this programme. Both designated and dedicated CHWs did, however, believe that additional training and more opportunities for counselling skills rehearsal would enhance the quality of counselling. Based on this feedback, we have extended the training schedule to 5 days to incorporate additional opportunities for role play and rehearsal, added a step-by-step guide to each counselling session in the training manual, and we plan to integrate additional training opportunities into weekly supervision for both the dedicated and designated CHWs. While this weekly supervision provides opportunities for CHWs to develop confidence and competence in their new job role, there may be some challenges to the provision of supervision in usual care. We anticipate challenges such as negative attitudes towards supervision, lack of space to provide supervision and a lack of priority given to supervision within the context of other priorities. Finding a suitably qualified person to deliver psychosocial supervision and support could also be a systems-level barrier that will require consideration before scaling up counselling. Qualitative work with CHWs tasked with delivering this programme and their experiences of supervision may help elucidate these barriers to supervision.

Finally, we found that CHWs require substantial support to overcome barriers to counselling implementation in chronic disease services. In this study, CHWs reported counselling and space challenges. Where chronic disease care teams were more supportive of mental health counselling, they created an enabling, therapeutic environment that facilitated counselling implementation. This is not altogether surprising given theories of implementation[31] and prior research in PHC services[32–34] that highlight contextual and organisational factors (such as leadership and climate) as critical drivers of counselling implementation. Given that PHC facilities are likely heterogenous with regards to readiness to implement mental health counselling, the start-up phase of a future trial will include facility readiness workshops aimed at ensuring relevant stakeholders are aware of and willing to support mental health counselling implementation. Although these barriers were raised by both dedicated and designated CHWs, on a whole, the issue was more salient for designated CHWs as they were managing

multiple expectations from the chronic disease care team. Designated CHWs had the additional constraint of managing their current workload in addition to this new service. Where designated CHWs were supported by their peers (who assisted with some of their usual tasks), they managed the additional responsibilities of delivering this intervention better. Based on this finding, we have developed a protocol for engaging with NGOs who employ designated CHWs that includes discussions about restructuring some of their usual HIV care responsibilities to accommodate mental health counselling activities.

Findings should be interpreted in light of some limitations. First, we gathered limited information on patients who refused screening and cannot determine if there were patient characteristics that distinguished those who declined and those who accepted screening. This should be addressed in future studies. Second, as there were only a small number of clusters in this feasibility test, we cannot draw inferences from the quantitative data. Similarly, the designated CHWs responsible for intervention delivery are probably not representative of the total population of CHWs working in chronic disease care. Third, our assessment of counselling fidelity was limited to a simple fidelity checklist that was not sensitive enough to detect more nuanced differences in counselling quality between dedicated and designated counsellors. As potential differences in counselling quality may help inform decisions about which model of care to invest in, we have developed a more comprehensive assessment of counselling fidelity for use in future studies.[22] Finally, as the study was based in PHC facilities in the Western Cape where resourcing for healthcare is somewhat better than other provinces, findings may not be generalisable to other parts of the country.

## CONCLUSION

Findings suggest that while it is largely feasible and acceptable to use either a dedicated or a designated approach to CHW-delivered mental health counselling, a few modifications to the recruitment, CHW training and counselling implementation protocols may enhance the likelihood of successful implementation. We have adjusted these protocols which are now being used in a cluster randomised controlled trial comparing the clinical and cost-effectiveness of a dedicated approach and a designated approach to CHW-delivered mental health counselling for improving the mental health and chronic disease outcomes of patients.[22]

## Author affiliations
[1]Alcohol, Tobacco and Other Drug Research Unit, South African Medical Research Council, Cape Town, South Africa
[2]Department of Psychiatry and Mental Health, University of Cape Town, Cape Town, South Africa
[3]Alan J Flisher Centre for Public Mental Health, Department of Psychiatry and Mental Health, University of Cape Town, Cape Town, South Africa
[4]Centre for Global Mental Health, Institute of Psychiatry, Psychology and Neuroscience, King's College, London, UK
[5]Department of Biostatistics Unit, South African Medical Research Council, Tygerberg, South Africa
[6]Division for Diabetes and Endocrinology, Department of Medicine, University of Cape Town, cape town, South Africa
[7]Nuffield Department of Primary Care Health Services, Oxford University, Oxford, UK
[8]Western Cape Department of Health, Cape Town, South Africa
[9]South African Medical Research Unit on Anxiety and Stress Disorders, Cape Town, South Africa

**Acknowledgements** We thank all study participants, participating facilities, NGOs, our stakeholder advisory group and our field team.

**Contributors** BM and KS conceived the project, performed the analyses and drafted the manuscript. CL, TN, CJL, NSL, JAJ, PM, CB and DJS helped develop and refine the project, including data tools, and revised the draft versions of the manuscript critically. PPW and CvdW played major roles in developing and organising the project, data collection and analyses, and revised draft versions of the manuscript critically. All authors read and approved the final manuscript.

**Funding** This study was supported by the joint-funded initiatives of the British Medical Research Council, Wellcome Trust and Department for International Development (MR/M014290/1) as well as funding from the South African Medical Research Council.

**Competing interests** None declared.

**Patient consent for publication** Not required.

**Provenance and peer review** Not commissioned; externally peer reviewed.

**Data sharing statement** Quantitative data are available on reasonable request from the corresponding author.

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
