## [Reviewer comments · BMJ Open]

ARTICLE DETAILS

TITLE (PROVISIONAL)	Community health worker-delivered counselling for common mental disorders among chronic disease patients in South Africa: a feasibility study
AUTHORS	Myers, Bronwyn; Petersen-Williams, Petal; van der westhuizen, Claire; Lund, Crick; Lombard, Carl; Joska, John; Levitt, Naomi; Butler, C; Naledi, Tracey; Milligan, Peter; Stein, Dan; Sorsdahl, Katherine

VERSION 1 – REVIEW

REVIEWER	Miya Barnett University of California, Santa Barbara, USA
REVIEW RETURNED	09-Jun-2018

GENERAL COMMENTS	The article "Community health worker-delivered counseling for common mental disorders among chronic disease patients in South Africa: a feasibility study" was well written and has the potential to make an important contribution to the literature. Most importantly, the study looks at dedicated and designated CHWs, which is important because a number of implementation questions exist regarding how to best mobilize CHWs to address mental health disorders and integrate them within care. 1. It would be helpful to address the following questions to better understand the sample: a. What was the gender of CHWs? Given that men were less likely to participate in counseling, it could be interesting to reflect on how the CHW gender and other factors (e.g., mental health stigma) impacted participation. b. Were there any differences in fidelity between the designated and dedicated CHWs? This could be an important area for investigation given that designated CHWs have additional competing interventions they need to learn and deliver. 2. It would be helpful to ground the discussion in the literature from implementation science, as many of the findings (e.g., the need for organizational support for CHW-delivered counseling intervention) are consistent with this area of study. 3. There were a few minor edits: a. In the abstract, the design section states "designated" approach twice as opposed to "designated and dedicated" b. The term "coloured" (Table 1) has different connotations outside of South Africa. Even with the footnote, it might be preferable to report on, "African only ancestry"
---

REVIEWER	Peter Ventevogel UNHCR, Switzerland
REVIEW RETURNED	21-Aug-2018

GENERAL COMMENTS	Review of Manuscript bmjopen-2018-024277 Community health worker-delivered counselling for common mental disorders among chronic disease patients in South Africa: a feasibility study General impression This is well written paper that focuses on the feasibility of a brief structured counselling intervention delivered by CHWs in South Africa. Specific attention is given to whether counselling in primary care is different when done by 'designated' CHW versus 'dedicated' CHW. This is a real issue and everyone who has set up a programme for mental health integration into primary care will have struggled with the question where to train designated or dedicated staff (clinicians or CHWs). This study does not provide the definite answers, as it is only a feasibility pilot, but it certainly contributes new elements to the knowledge pool around task shifting approaches in mental health in sub-Saharan Africa. Major suggestions I do not have major comments or suggestions. Minor suggestions  1. P 10 line 29. Please check figures. It can now be understood as if 95% of the patients in rural areas refuse screening while only 5% in urban refuges. Is that what the authors want to say? If not, please correct this, and if yes, this is a major limitation of scaling up that should be discussed later in the paper. 2. Perhaps the authors can refer to work of others in South Africa and elsewhere. See for example the work of van de Water, Rossouw, Yadin, and Seedat (2018) and of Jerene et al. (2017) although admittedly these papers focus on different interventions. 3. A critical element in task-sharing approaches is the availability of supportive clinical supervision. That is often the crux of a good programme as it is during supervision sessions that significant learning will take place and confidence is being built. Good supervision makes it possible to limit the classroom-based trainings to just a few days. Can the authors say a bit more on how they organized clinical supervision in thss project (it is now just one line on p 9. line 31-35) and what challenges can be expected when expanding the intervention
---

	from a research setting involving a university department to routine healthcare settings in a rural subsaharan context? 4. The issue of low interest of men with alcohol use problems to participate in counselling can be highlighted stronger in the discussion as it is a real issue globally and certainly also in Sub Sharan Africa. See e.g. Tol et al. (2018). Perhaps the need to adapt interventions to make them more attractive to men can be discussed more explicitly in the last sections of the paper. 5. The discussion could come back a bit more specifically on the difference between 'dedicated' and 'designated' CHWs, something that features prominently in the abstract but does not get much attention in the discussion. References  • Jerene, D., Biru, M., Teklu, A., Rehman, T., Ruff, A., & Wissow, L. (2017). Factors promoting and inhibiting sustained impact of a mental health task-shifting program for HIV providers in Ethiopia. Global Mental Health, 4. • Tol, W., Augustinavicius, J., Carswell, K., Brown, F., Adaku, A., Leku, M., . . . Van Ommeren, M. (2018). Translation, adaptation, and pilot of a guided self-help intervention to reduce psychological distress in South Sudanese refugees in Uganda. Global Mental Health, 5. • van de Water, T., Rossouw, J., Yadin, E., & Seedat, S. (2018). Adolescent and nurse perspectives of psychotherapeutic interventions for PTSD delivered through task-shifting in a low resource setting. PloS One, 13(7), e0199816.
--	--

VERSION 1 – AUTHOR RESPONSE

Reviewer: 1

1. It would be helpful to address the following questions to better understand the sample:

a. What was the gender of CHWs? Given that men were less likely to participate in counseling, it could be interesting to reflect on how the CHW gender and other factors (e.g., mental health stigma) impacted participation.

This is an important consideration. We have added this information to the discussion and in context of the lower rates of male participation.

b. Were there any differences in fidelity between the designated and dedicated CHWs? This could be an important area for investigation given that designated CHWs have additional competing interventions they need to learn and deliver.

We agree with you and although we didn't detect differences in this small pilot, we are systematically assessing for differences in the trial and this will be an important consideration about how to proceed with integration. We have added this to the limitations section

2. It would be helpful to ground the discussion in the literature from implementation science, as many of the findings (e.g., the need for organizational support for CHW-delivered counseling intervention) are consistent with this area of study.

Thank you- we have tried to do this and have added literature as appropriate

3. There were a few minor edits:

a. In the abstract, the design section states “designated” approach twice as opposed to “designated and dedicated” Thanks- we have corrected this

b. The term "coloured" (Table 1) has different connotations outside of South Africa. Even with the footnote, it might be preferable to report on, "African only ancestry" Thanks we have changed this

Reviewer: 2

P 10 line 29. Please check figures. It can now be understood as if 95% of the patients in rural areas refuse screening while only 5% in urban refuges. Is that what the authors want to say? If not, please correct this, and if yes, this is a major limitation of scaling up that should be discussed later in the paper. No this is a major error, we have re-run the analyses and corrected accordingly

2. Perhaps the authors can refer to work of others in South Africa and elsewhere. See for example the work of van de Water, Rossouw, Yadin, and Seedat (2018) and of Jerene et al. (2017) although admittedly these papers focus on different interventions. The reviewer is correct that these papers address important but different questions. We have noted however that this paper contributes to the body of work on task-sharing interventions in Africa (with appropriate citations)

3. A critical element in task-sharing approaches is the availability of supportive clinical supervision. That is often the crux of a good programme as it is during supervision sessions that significant learning will take place and confidence is being built. Good supervision makes it possible to limit the classroom-based trainings to just a few days . Can the authors say a bit more on how they organized clinical supervision in thss project (it is now just one line on p 9. line 31-35) and what challenges can be expected when expanding the intervention from a research setting involving a university department to routine healthcare settings in a rural subsaharan context? Thank you- we have added more detail to the methods and the discussion on clinical supervision.

4. The issue of low interest of men with alcohol use problems to participate in counselling can be highlighted stronger in the discussion as it is a real issue globally and certainly also in Sub Sharan Africa. See e.g. Tol et al. (2018). Perhaps the need to adapt interventions to make them more attractive to men can be discussed more explicitly in the last sections of the paper. We agree and have added some information about this. Reaching men is a major issue in the health system generally.

5. The discussion could come back a bit more specifically on the difference between ‘dedicated’ and ‘designated’ CHWs, something that features prominently in the abstract but does not get much attention in the discussion. On re-reading the discussion, we understand why this comment was made. We have tried to highlight the two models more in the discussion.

VERSION 2 – REVIEW

REVIEWER	Miya White Barnett University of California, Santa Barbara
REVIEW RETURNED	13-Oct-2018

GENERAL COMMENTS	The authors have been responsive to reviewer comments. Only minor edits are needed at this time:
--

	1. I found a few typos. a. In the abstract there needs to be a space between 7 and CHW b. Please correct this sentence, "ifted to registered psychological counsellors in line with the national mental health policy framework." on page 8. 2. I believe the sentence on page 14, which states, "Not only were there benefits for patients who received this intervention, but other patients also seemed to benefit from the designated CHWs' enhanced counselling skills," should say "perceived benefits" as this paper does not report on the effectiveness of the interventions.
--	--

REVIEWER	Peter Ventevogel UNHCR, Geneva
REVIEW RETURNED	21-Oct-2018

GENERAL COMMENTS	The authors made (minor) revision as was requested by the reviewers. The paper is ready to be accepted in my view, A small thing: in the the text under 'Description of counselling programmes and CHWs;' the following text is duplicated: ifted to registered psychological counsellors in line with the national mental health policy framework'.
---

VERSION 2 – AUTHOR RESPONSE

Thank you for the positive, thoughtful review of the revised manuscript. We appreciate the opportunity to strengthen the manuscript even further and have made the proposed edits.

- a. In the abstract there needs to be a space between 7 and CHW -this has been corrected
- b. Please correct this sentence, "ifted to registered psychological counsellors in line with the national mental health policy framework." on page 8. This change has been made
2. I believe the sentence on page 14, which states, "Not only were there benefits for patients who received this intervention, but other patients also seemed to benefit from the designated CHWs' enhanced counselling skills," should say "perceived benefits" as this paper does not report on the effectiveness of the interventions. We agree and have added the word perceived to clarify this